# Aerosol forcing regulating recent decadal change of summer water vapor budget over the Tibetan Plateau

Zhili Wang [1,3] ✉, Yadong Lei[1,3], Huizheng Che [1] ✉, Bo Wu[2] & Xiaoye Zhang[1]

The Tibetan Plateau (TP), known as the Asian water tower, has been getting wetter since the 1970s. However, the primary drivers behind this phenomenon are still highly controversial. Here, we isolate the impacts of greenhouse gases (GHG), aerosols, natural forcings and internal climate variability on the decadal change of summer water vapor budget (WVB) over the TP using multi-model ensemble simulations. We show that an anomalous Rossby wave train in the upper troposphere travelling eastward from central Europe and equatorward temperature gradient in eastern China due to the inhomogeneous aerosol forcing in Eurasia jointly contribute to anomalous easterly winds over the eastern TP. Such anomalous easterly winds result in a significant decrease in water vapor export from the eastern boundary of the TP and dominate the enhanced summer WVB over the TP during 1979-2014. Our results highlight that spatial variation of aerosol forcing can be used as an important indicator to project future WVB over the TP.

The Tibet Plateau (TP) and its surrounding high mountain areas are the second largest glacier-gathering area in the world except for polar ice sheets, storing a large amount of water resources in the form of glaciers, snow, lakes, and rivers[1–3]. Meanwhile, the TP is the birthplace of twelve important rivers in Asia, such as the Yangtze River, the Yellow River, the Yarlung Zangbo River, the Indus River, and the Ganges River. It provides necessary water resources for almost two billion people, and is known as the Asian water tower[3–5]. Therefore, change in water resources over the TP will seriously affect the natural ecosystems and socio-economic development in the TP itself and its surrounding countries.

The unique high-altitude terrain and atmospheric circulation dominated by monsoon and upper westerly bring abundant water resources to the TP[6,7]. However, climate change has greatly affected the hydroclimatic changes in the TP[1,8–10]. In the past decades, the TP has shown significant warming and humidification[3,11–15]. Meanwhile, the average precipitation over the TP has increased, mainly manifested as an increase in the northwest but a decrease in the southeast[16,17]. This

phenomenon and the mechanism behind it have received widespread concerns from the scientific community and policymakers.

As one of the most active components of the water cycle, atmospheric water vapor transport determines the spatial pattern and trends of water resources in the Asian water tower[4,18]. Summer is the season with the highest water vapor budget (WVB) over the TP[19]. Previous studies reported a significant increasing trend in summer net WVB over the TP (especially in the north) since the late 1970s[4,20]. However, the dominant factors of increase in WVB over the TP are still highly controversial. On the one hand, enhanced atmospheric water content capacity driven by global warming may lead to the increase of water vapor content over the TP[4,17]. On the other hand, changes in water vapor transport driven by large-scale circulation are identified to be the main drivers of the interdecadal changes in WVB over the TP[3]. The weakening of the mid-latitude westerlies in the upper troposphere in Eurasia and the resulting decrease in water vapor export from the eastern boundary of the TP may dominate the interdecadal increase in net WVB over the TP[20,21]. Oceanic internal variability such as changes in

[1]State Key Laboratory of Severe Weather & Key Laboratory of Atmospheric Chemistry of CMA, Chinese Academy of Meteorological Sciences, Beijing 100081, China. [2]State Key Laboratory of Numerical Modeling for Atmospheric Sciences and Geophysical Fluid Dynamics (LASG), Institute of Atmospheric Physics, Chinese Academy of Sciences, Beijing 100029, China. [3]These authors contributed equally: Zhili Wang, Yadong Lei. ✉e-mail: wangzl@cma.gov.cn; chehz@cma.gov.cn

sea surface temperatures in North Atlantic and Pacific may partially explain the westerly jet anomalies and changes in WVB in the northern TP[16,20]. However, an early study suggested that the high Asian mountains prevented further propagation of precipitation minus evaporation (PME) deficit originating from the southeastern North Atlantic to the central TP region, which in turn may have contributed to the wetting of north-central TP[22].

As the second largest climate forcing factor, aerosol forcing has played a crucial role in regional climate change over the industrial era[23–27]. A recent study showed that the inhomogeneous changes in anthropogenic aerosol emissions in Eurasia since the 1970s (i.e., decreasing in Europe and increasing in Asia) was likely the main driver of the interdecadal weakening of the Eurasian subtropical westerly jet during summer[28]. This study raises the question of whether aerosol forcing is linked to the decadal changes of summer WVB over the TP.

Here we examine the changes of summer WVB over the TP during the past four decades (1979–2014) using a climate reanalysis dataset. We disentangle the contributions of greenhouse gases (GHG), aerosols, natural forcings, and internal climate variability to the decadal trend of summer WVB over the TP using the multi-model simulations from the Coupled Model Intercomparison Project Phase 6 (CMIP6) (Supplementary Table 1), and identify the dominant role of aerosol forcing. Our results provide valuable insights into the mechanism of change in WVB over the TP.

## Results

### Observed changes in WVB over the TP

The water vapor sources over the TP mainly come from the water vapor transports of mid-latitude westerly winds in the upper troposphere in northern Hemisphere and Asian monsoon[6,29]. An amount of moisture is carried into the TP from its western, northern and southern boundaries by the westerly winds and Indian monsoon, respectively, while part of it is exported from its eastern boundary (Supplementary Fig. 1a). The ERA5 reanalysis shows that the summertime WVB (represented by the difference between precipitation and evaporation) over the entire TP is increasing significantly in the past four decades (1979–2014), especially in the western and northern regions, where the maximum increased rate exceeds 5 mm month⁻¹ decade⁻¹ (Fig. 1a). The increase of WVB has greatly contributed to the wetting trend of the TP in recent decades[22,30,31] (Supplementary Fig. 2c),

although there is a slight increase in evaporation due to global warming[32] (Supplementary Fig. 2b).

The increasing summer WVB over the TP can be attributed to the changes of atmospheric moisture storage and moisture flux divergence (Details in Eq. 1 in "Method" section). During 1979–2014, the vertically integrated moisture flux divergence shows a significant increase ($p < 0.1$), with a seasonal average rate of 4.7 kg s⁻¹ decade⁻¹. In contrast, there are limited changes of atmospheric moisture storage, with a seasonal average rate of −0.05 kg s⁻¹ decade⁻¹. This comparison demonstrates the dominated contribution of moisture flux divergence to increasing WVB over the TP. Furthermore, we decompose the changes of vertically integrated moisture flux divergence to the changes of water vapor fluxes in four boundaries (Details in Eqs. 2–3 in "Method" section). The results show that there is a significant positive trend of $6 \times 10^6$ kg s⁻¹ decade⁻¹ ($p < 0.1$) in summer water vapor flux in the eastern boundary of the TP during 1979–2014 (Fig. 1b), indicating a marked decrease of water vapor export from this boundary. However, the summer water vapor input from the northern boundary significantly decrease at a rate of $2.8 \times 10^6$ kg s⁻¹ decade⁻¹ ($p < 0.1$) during 1979–2014. There are limited changes in water vapor inputs (not passing statistical significance at 90% confidence level) from the western and southern boundaries of the TP. Therefore, the decrease in water vapor export from the eastern boundary dominates the increase in net WVB over the TP in summer during 1979–2014.

### Attribution of changes in water vapor flux in eastern boundary of the TP

The changes of summer WVB over the TP in the last four decades are likely influenced by multiple external forcings and internal climate variability[33,34]. Isolating these influences can only depends on climate models. The CMIP6 multi-model ensemble (MME) results show that the historical all forcing (ALL) simulations can reproduce the observed positive trend of water vapor flux in the eastern boundary of the TP in the past four decades (Supplementary Table 2). Notably, the CMIP6 models underestimate the increase of summer water vapor flux in the eastern boundary of the TP, with a MME averaged trend of 0.8 kg s⁻¹ decade⁻¹. This could be primarily related to the systematic underestimation of climatological vertically integrated water vapor and WVB over the TP and its surrounding areas by the climate models[34,35] (Supplementary Fig. 1b vs. 1a).

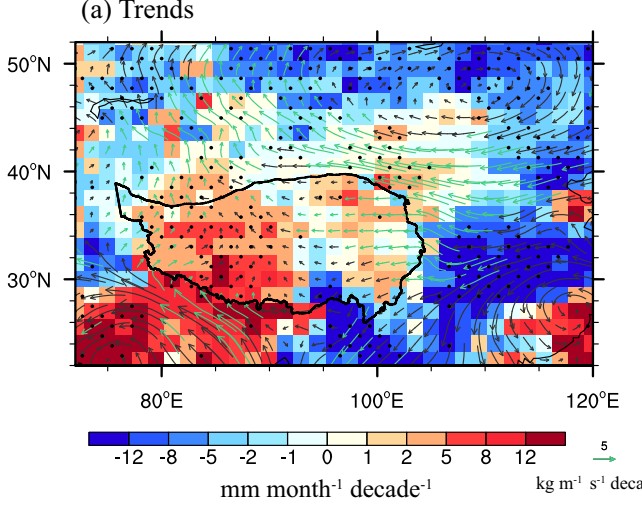

(a) Trends

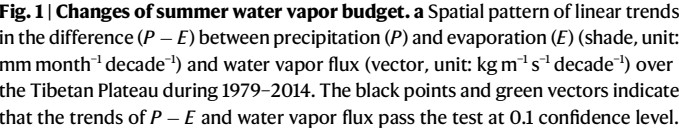

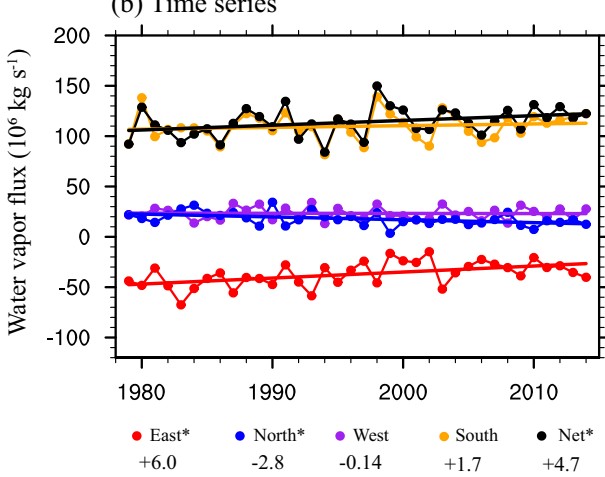

(b) Time series

**Fig. 1 | Changes of summer water vapor budget. a** Spatial pattern of linear trends in the difference ($P − E$) between precipitation ($P$) and evaporation ($E$) (shade, unit: mm month⁻¹ decade⁻¹) and water vapor flux (vector, unit: kg m⁻¹ s⁻¹ decade⁻¹) over the Tibetan Plateau during 1979–2014. The black points and green vectors indicate that the trends of $P − E$ and water vapor flux pass the test at 0.1 confidence level.

**b** Time series (point) and linear fitting (line) of water vapor flux (unit: 10⁶ kg s⁻¹) in four boundaries of the Tibetan Plateau during 1979–2014. The asterisks represent statistical significance at 90% confidence level. The numbers represent linear trends (unit: 10⁶ kg s⁻¹ decade⁻¹) of water vapor flux in each boundary of the Tibetan Plateau.

Furthermore, we quantify the impacts of multiple external forcings and internal climate variability on the enhanced summer water vapor flux in the eastern boundary of the TP during 1979–2014 using ALL, single-forcing (AER, GHG, and NAT) and piControl simulations (Fig. 2). Model results reveal that the aerosol forcing dominates the enhanced water vapor flux in the eastern boundary of the TP in the past four decades (red point in Fig. 2) in the historical all forcing simulations, with trends of 1.1 vs. 0.8 kg s$^{-1}$ decade$^{-1}$ in the MME. In addition, internal climate variability also contributes to a wetter TP (orange point in Fig. 2), but the value is less than half of that from aerosol forcing. In a contrary, greenhouse gas and natural forcings show a negative contribution to the enhanced water vapor flux in eastern boundary of the TP (blue and green points in Fig. 2), which partly offset the impacts of aerosol forcing and internal climate variability. Our attribution analysis is generally consistent with two earlier studies[33,34], which conclude that the recent decadal changes of WVB over the TP

are mainly attributed to combined effects of anthropogenic forcings and internal climate variability. Nonetheless, our study further isolates the impacts of greenhouse gas and aerosol forcings and overweight the dominated contribution from aerosol forcing rather than internal climate variability to enhanced WVB over the TP.

**Potential mechanisms**

The changes of water vapor flux are determined by both wind speed and specific humidity, representing dynamic and thermodynamic processes, respectively. By fixing specific humidity (wind speed) and varying wind speed (specific humidity), we decompose anomalous summer water vapor flux in the eastern boundary of the TP to the contributions from the changes in dynamic and thermodynamic processes (Fig. 3). It is seen that the changes of zonal wind in the upper troposphere (i.e., anomalous easterly winds) dominate the decrease in summer water vapor export from the eastern boundary during 1979–2014 (red bar vs. gray bar in Fig. 3), which primarily contributes to the increase in summer WVB over the TP. In a contrary, the changes of specific humidity slightly decrease the summer water vapor flux in the eastern boundary of the TP (blue bar in Fig. 3a). The dominated contribution from dynamic process to enhanced summer water vapor flux in the eastern boundary of the TP is also captured by climate models from CMIP6, despite differences in amplitude (Fig. 3b).

By calculating the linear trends of global summer winds at 500 hPa, we further examine the possible mechanisms of the interdecadal change in anomalous easterly winds driven by external forcings, such as greenhouse gases and aerosols (Fig. 4). Observation shows an anomalous wave train in the upper troposphere propagating eastward from central Europe to East Asia during 1979–2014 (Supplementary Fig. 3). This wave train leads to significant positive anomalies in geopotential height at 500 hPa near Lake Baikal, indicating an anomalous anticyclone here. Anomalous easterly winds have increased markedly in the south of the anomalous anticyclone, which contribute to the decrease in water vapor export from the eastern boundary of the TP caused by the climatological westerly winds and increase in water vapor over the TP. Such anomalous anticyclone near Lake Baikal and easterly winds over the TP are clear in ALL and AER simulations (Fig. 4a, c), but not seen in GHG and NAT simulations (Fig. 4b, d). The anomalous wave train in the upper troposphere in Eurasia is also reproduced in AER simulation (Fig. 4c), highlighting the leading role of aerosols in regulating interdecadal change in easterly anomalies.

In the past decades, the anthropogenic aerosol emissions in Europe show large decline to meet the European Union's target for human

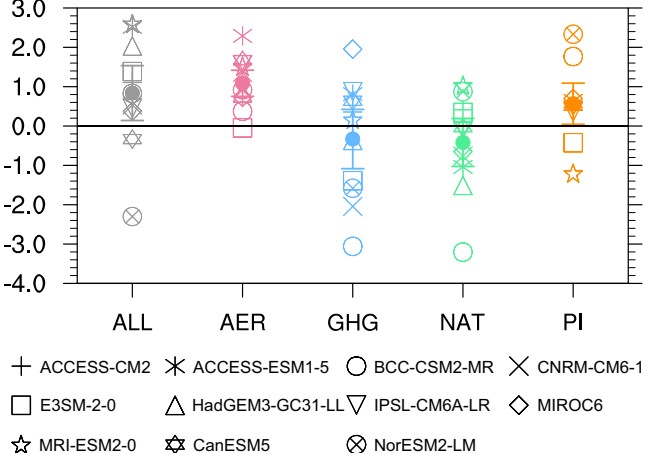

**Fig. 2 | Attribution of changes in summer water vapor flux in the eastern boundary of the Tibetan Plateau.** The gray, red, blue, green, and orange markers represent simulated linear trends (unit: 10$^6$ kg s$^{-1}$ decade$^{-1}$) of water vapor flux in eastern boundary of the Tibetan Plateau during 1979–2014 from all (ALL), aerosol-only (AER), greenhouse gas-only (GHG), natural-only (NAT) forcings and piControl (PI) simulations in eleven climate models, respectively. The points represent the ensemble means of eleven climate models. The error bars represent one standard deviation of eleven climate models.

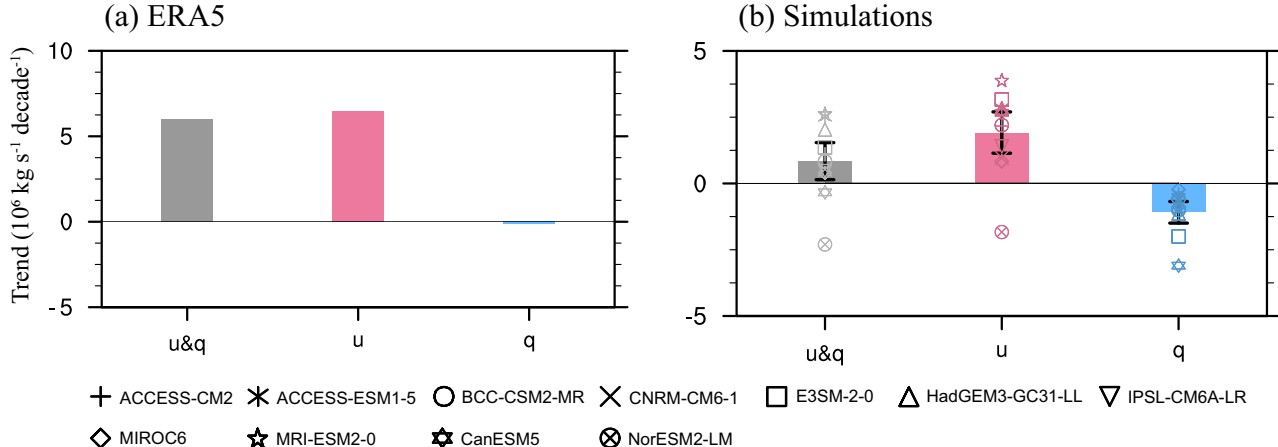

**Fig. 3 | Linear trends of water vapor flux.** Observed (**a**) and simulated (**b**) linear trends (unit: 10$^6$ kg s$^{-1}$ decade$^{-1}$) of water vapor flux (gray bars) in eastern boundary of the Tibetan Plateau and its contributions from changes of zonal wind (red bars) and specific humidity (blue bars) during 1979–2014. The markers in panel (**b**)

represents the simulated values in eleven climate models. The bars and error bars in panel (**b**) represent ensemble means and one standard deviations of eleven climate models.

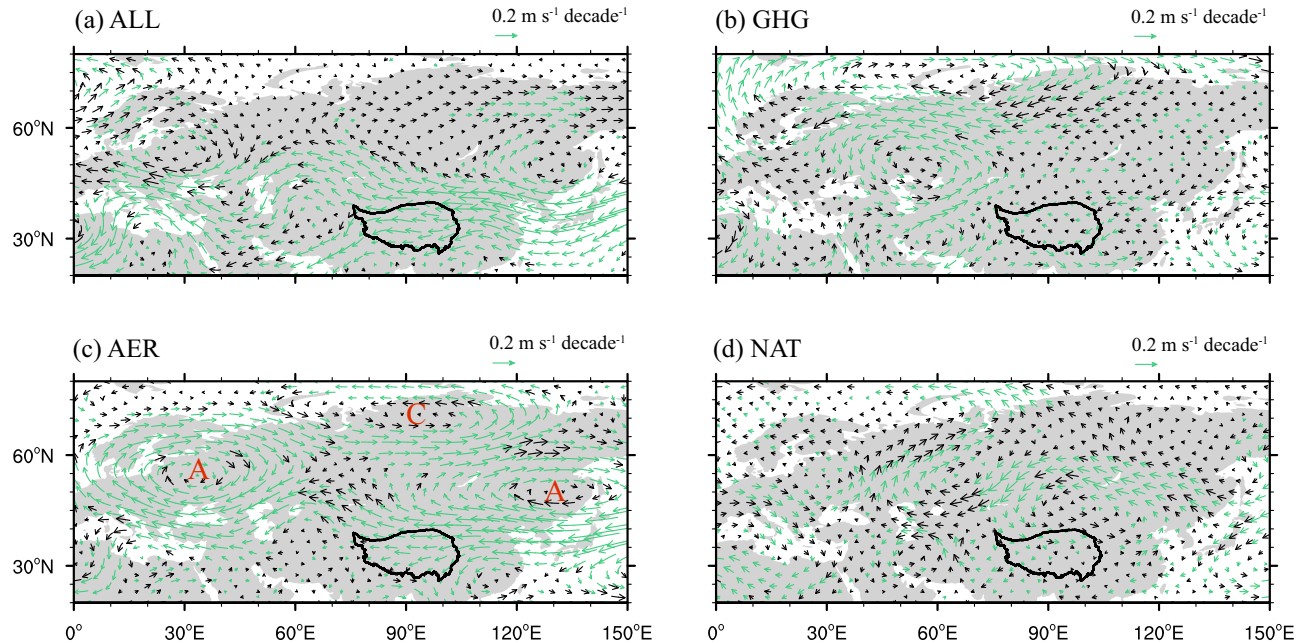

**Fig. 4 | Linear trends of summer wind vector.** The linear trends of summer mean wind vector (unit: m s⁻¹) at 500 hPa during 1979–2014 in all (ALL, **a**), greenhouse gas (GHG, **b**), aerosol (AER, **c**), and natural (NAT, **d**) forcing simulations from CMIP6. The green vectors represent changes with high inter-model agreement defined as at least eight of eleven models agreeing on the direction of change.

health[36,37]. From 1979 to 2014, the aerosol optical depth (AOD) in Europe (0–40°E, 40–60°N) decreases at a rate of 0.07 per decade in ALL simulation (Fig. 5a), causing a large disturbance on regional climate change[26,38]. The reduced AOD greatly increase the surface solar radiation (SSR) (Fig. 5b), resulting in a local warming in the lower atmosphere (Fig. 5c). The diabatic heating can induce anomalous ascending motion and then excite an anomalous Rossby wave train in the upper troposphere, which propagates energy eastward from Europe (Fig. 4). As a consequence, an anomalous anticyclone circulation at 500 hPa is formed near Lake Baikal, which continuously weakens the westerlies in the eastern boundary of the TP, leading to weakened water vapor export from the eastern boundary of the TP in the past four decades. Therefore, the decrease in water vapor export from the eastern boundary driven by atmospheric circulation anomaly dominates the increase in summer precipitation over the TP. The more latent heat release from increasing precipitation may strengthen convection development, which may in turn intensify anomalous anticyclone in the upper troposphere of the northern TP[39], thus forming a local positive feedback effect. Such an anomalous Rossby wave train driven by aerosol reductions in Europe was also reported in an early study[40].

Additionally, the anomalous anticyclone circulation can cause local warming in the upper-middle troposphere northeastern of TP (Fig. 5d). Meanwhile, a local cooling is found to the east of TP, which is mainly attributed to increased aerosol emissions in eastern China during 1979–2014 (Fig. 5a). Thus, an anomalous temperature gradient towards equator forms in the upper troposphere over the eastern China (Supplementary Fig. 4a), which further weakens the westerly winds in the eastern boundary of the TP based on the thermal wind relationship (Supplementary Fig. 4b). Such an anomalous temperature gradient and weakening westerly over TP are reproduced again in ALL and AER simulations (Fig. 6a, b, e, f). In a contrary, the westerly over the TP is increased slightly in NAT simulations due to the enhanced poleward temperature gradient (Fig. 6g, h), posing a negative effect on enhanced water vapor flux in the eastern boundary of the TP in the past few decades. Noticeably, although slightly weakened westerlies in the eastern boundary of the TP due to the anomalous temperature

gradient towards equator is shown in GHG simulations (Fig. 6c, d), GHG forcing also shows a negative effect on enhanced water vapor flux in the eastern boundary of the TP (Fig. 2). Such discrepancy is attributed to the impact of thermodynamic process rather than dynamic process in GHG simulations (Supplementary Fig. 5).

## Discussion

This study provides a comprehensive analysis of the driving forces of decadal change in WVB over the TP during 1979–2014 based on the CMIP6 MME simulations and ERA5 reanalysis dataset. The observation shows that the anomalous easterly winds in the upper troposphere decreases the water vapor export from the eastern boundary, which dominates the increase in WVB over the TP. Analysis of climate model simulations indicates that the aerosol forcing dominates the decrease in water vapor export from the eastern boundary of the TP. Our results further reveal two impact paths driven by the inhomogeneous aerosol forcing in Eurasia in the past four decades (Fig. 7). Anomalous Rossby wave train in the upper troposphere traveling eastward across Eurasia due to the aerosol decline in Europe and anomalous temperature gradient towards equator due to the aerosol rise in eastern China jointly contribute to anomalous easterly winds over the eastern TP and enhanced water vapor over the TP. Our study links the WVB change over the TP with aerosol forcing and provides valuable insights into the mechanism of recent increase in WVB over the TP.

Our findings that the decrease in water vapor export from the eastern boundary dominates the increase in net WVB over the TP in summer are generally agreed with Zhou et al.[20], but don't support the suggestion of Zhang et al[22]. that the increase in water vapor over the TP was attributable to the prevention of high Asian mountains for propagation of PME deficit. Meanwhile, compared with a recent report by Jiang et al.[41], which showed that the decrease in meridional troposphere temperature gradient over Eurasia caused by the uneven anthropogenic aerosol emissions led to the weakening of the Asian subtropical westerly jet, which ultimately dominated the increase in summer precipitation in the northern TP since the 1950s, we suggests a new mechanism, i.e., a teleconnection pathway across Eurasia induced by European aerosol forcing. We further reveal that the above

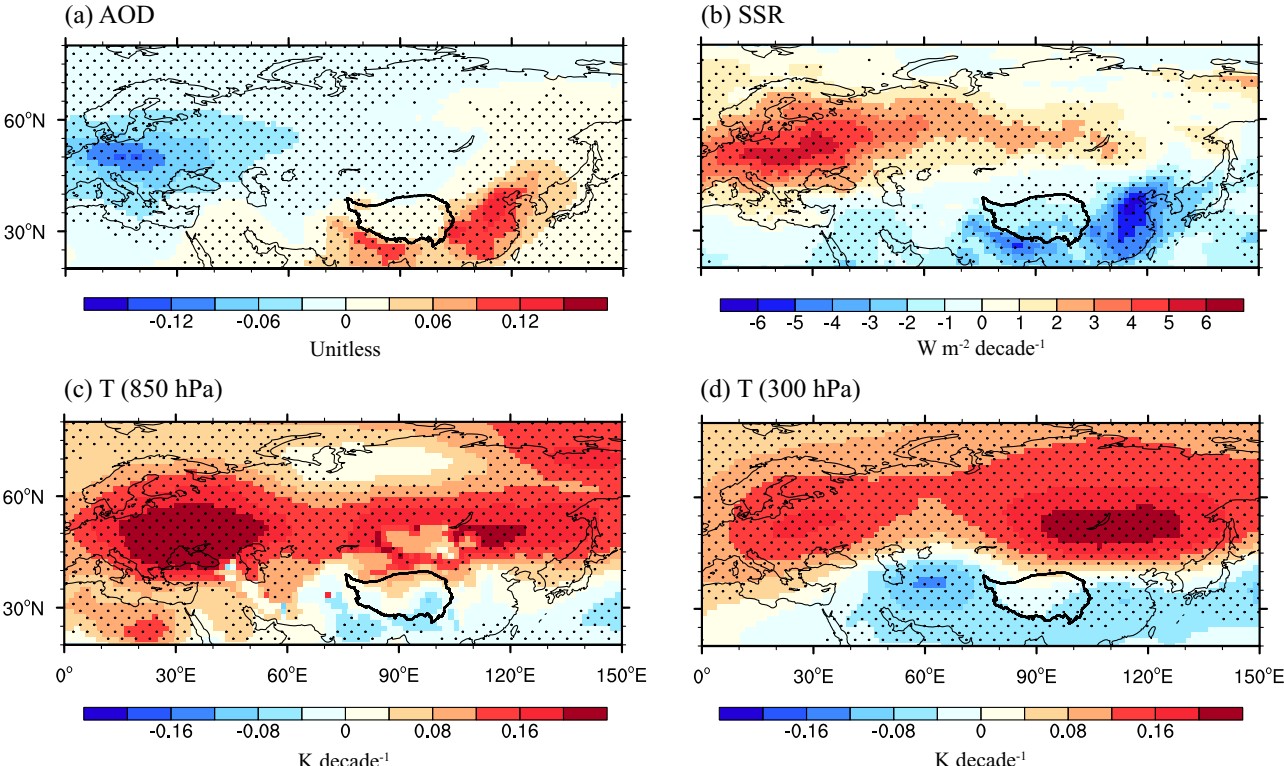

**Fig. 5 | Linear trends of summer aerosol optical depth (AOD), all-sky surface solar radiation (SSR), and temperature (T). a, b** The linear trends of summer mean AOD (unit: unitless) in all forcing simulation and SSR (unit: W m⁻² decade⁻¹) in aerosol-only simulation during 1979–2014. **c, d** The linear trends of temperature (unit: K decade⁻¹) at 850 hPa and 300 hPa in aerosol-only simulation during 1979–2014. Black points represent changes with high inter-model agreement defined as at least eight of eleven models agreeing on the direction of change.

mechanism and anomalous temperature gradient in eastern China due to the uneven aerosol forcing in Eurasia collectively play a dominant role in the weakening of the westerly jet over the eastern TP. The weakened westerly winds decrease the water vapor output from the eastern boundary of the TP, rather than other boundaries, which may be the primary cause of increase in precipitation in the northern TP.

In this study, we use model simulations from the CMIP6 to attribute the recent decadal change of summer WVB over the TP. There are two aspects worth noting here: (i) Although the only first three ensemble members in each model are used in this study, the main conclusion that aerosol forcing regulate the recent decadal change of summer water vapor budget over the Tibetan Plateau remains the same in ten ensemble members from IPSL-CM6A-LR and MIROC6 (Supplementary Fig. 6). (ii) Although climate models can reproduce the enhanced WVB over the TP in the past four decades, we must acknowledge that climate models underestimate the wetting trend of the TP, which is also reported in previous studies[16,35]. This underestimation may be attributed to the following aspects: (i) Climate models systematically underestimate the climatological WVB over the TP[16,35], may lead to the weakened response of WVB over the TP to anthropogenic forcings. (ii) Large biases in emissions inventory as adopted by the CMIP6 models may contribute to underestimated response of WVB over the TP to aerosol forcing[40,42]. However, such underestimation in model simulations may cause limited impacts on our attribution analysis, because AER simulation can account for 130% of enhanced summer WVB over the TP in ALL simulation.

Our study highlights the dominated contribution from the inhomogeneous changes in anthropogenic aerosol emissions over Eurasia to regulating the decadal increase of summer WVB over the TP in the past four decades. However, the trend in WVB over the TP, especially in its eastern part, will probably be changed with further variation in spatial pattern of aerosol emissions. In fact, such situation has already occurred for more than a decade (Supplementary Fig. 7a). Since around 2010, the AOD has shown a limited change in Europe but sharp decline in China caused by stringent clean air actions implemented by Chinese government (Supplementary Fig. 7b). As a consequence, the enhanced SSR (Supplementary Fig. 7c, d) caused by the air pollution mitigation may exacerbate the atmospheric warming and contribute to an anomalous northward temperature gradient in eastern China (Supplementary Fig. 8a). The increased poleward temperature gradient can enhance the upper-tropospheric westerlies in the eastern TP and its downstream areas (Supplementary Fig. 8b), which may contribute to increase in water vapor export from the eastern boundary of the TP and decrease in net WVB over the TP during summer (Supplementary Fig. 7a). However, the CMIP6 models fail to capture the trends in AOD and SSR in China during this time due to an opposite trend of anthropogenic aerosol emissions in the Community Emissions Data System (CEDS) inventory[42,43]. It will be worthwhile to quantify the impact of reductions in aerosol emissions over China in the future by using specific model experiments such as the recent Regional Aerosol Model Intercomparison Project (RAMIP)[44]. It is foreseeable that the water vapor export from the eastern boundary of the TP may continue to be enhanced in the near-term future due to concomitant reductions in aerosol emissions with deep decarbonization in China to meet the target of carbon neutrality by 2060[45]. Therefore, spatial variation of aerosol forcing needs to be taken into account to project the future water cycle over the TP.

## Methods
### Observation and model outputs
We choose the fifth-generation climate reanalysis of the European Centre for Medium-Range Weather Forecast (ERA5) as benchmark to

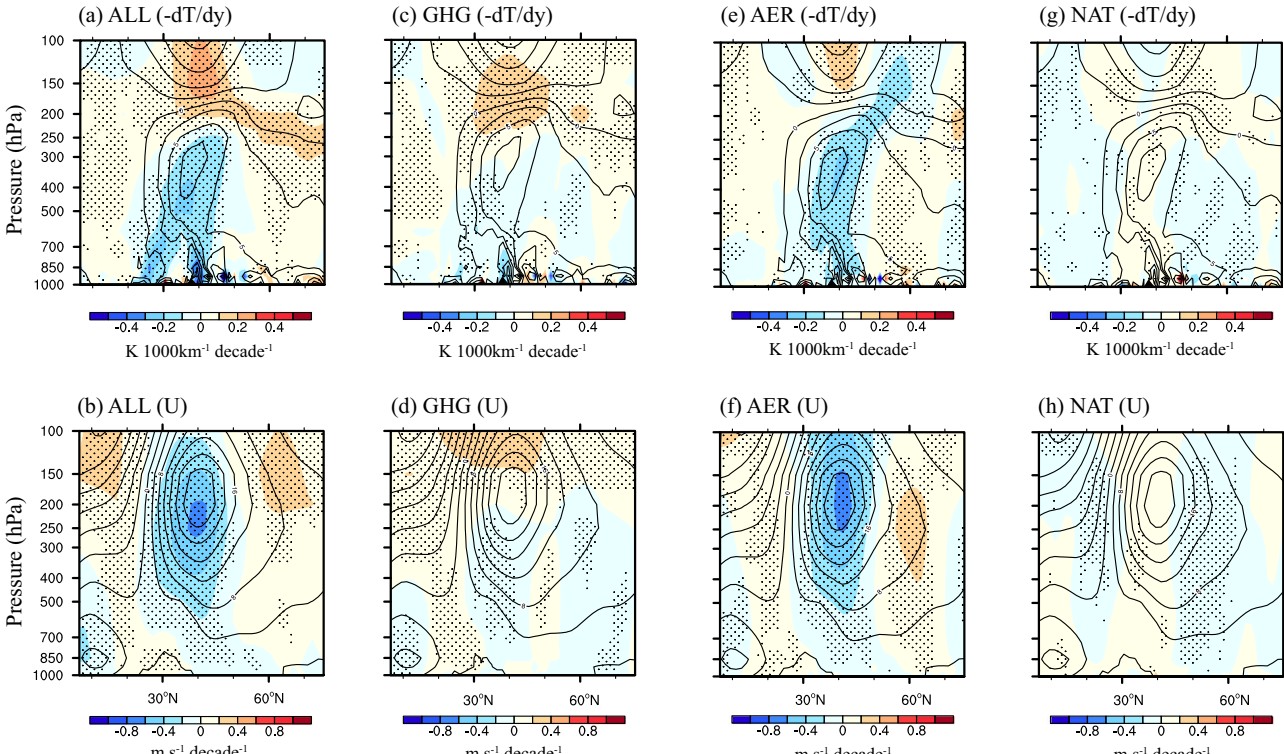

**Fig. 6 | Trends of summer meridional temperature gradient and zonal wind.** Climatology (contour) and linear trends (shade) of summer mean meridional temperature gradient (-dT/dy, unit: K 1000 km$^{-1}$ decade$^{-1}$) and zonal winds (unit: m s$^{-1}$ decade$^{-1}$) zonally averaged between 90°E and 120°E during 1979–2014 in all (ALL, **a**, **b**), greenhouse gas (GHG, **c**, **d**), aerosol (AER, **e**, **f**) and natural (NAT, **g**, **h**) forcing simulations from CMIP6. The black points indicate changes with high inter-model agreement defined as at least eight of eleven models agreeing on the direction of change.

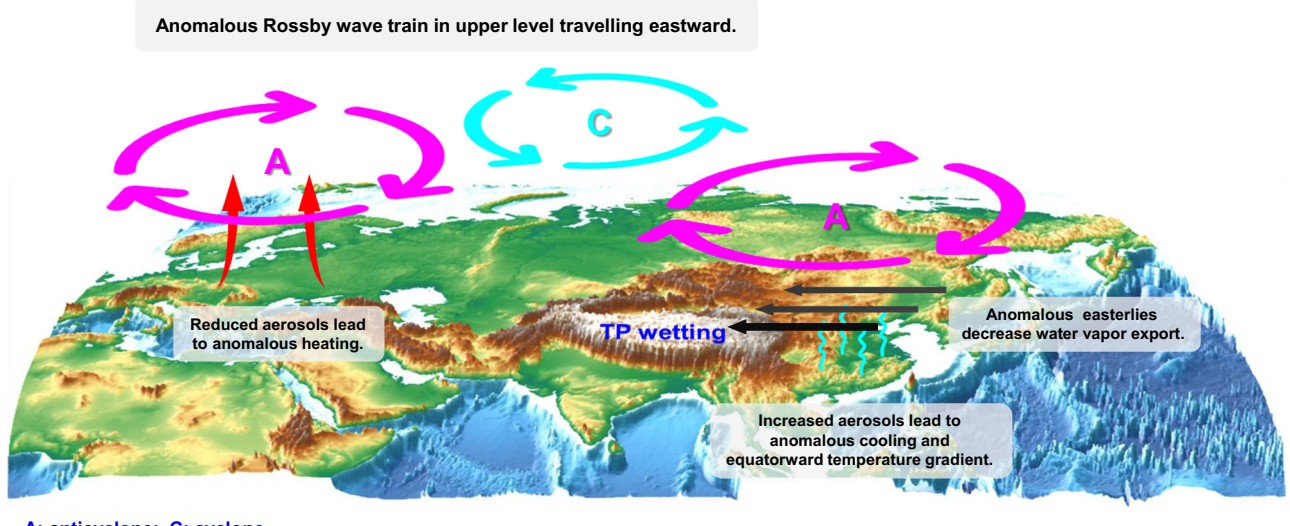

**Fig. 7 | Schematic diagram of physical mechanisms.** The pathways of inhomogeneous aerosol forcing in Eurasia regulating the recent decadal change of summer water vapor budget over the Tibetan Plateau.

evaluate the performance of climate models from the CMIP6 and quantify the impacts of various factors on the WVB, due to the lack of long-term continuous multivariate observations in the TP. Because of its high reliability, the ERA5 reanalysis has been widely used in climate change research across regions, including the TP[46–48]. Monthly variables, including specific humidity, zonal and meridional winds, surface pressure, and geopotential height, during 1979–2014 are used to investigate the decadal change of WVB in the TP.

Considering the availability of all required variables in this study, eleven climate models in CMIP6 are selected to quantify the contributions of anthropogenic and natural forcings to the decadal change of WVB over the TP (Supplementary Table 1). These models

provide both historical all forcings simulations (ALL) and single-forcing simulations, including GHG forcing only (GHG), anthropogenic aerosol forcing only (AER), and natural forcing only (NAT, solar, and volcanic combined) from the Detection and Attribution Model Intercomparison Project (DAMIP). For a specific scenario, the outputs of the first three ensemble member (r1i1p1f, r2i1p1f, and r3i1p1f) in each model are used in this study. Moreover, 44 chunks of nonoverlapping 36-year time series from preindustrial control simulations (piControl) with ten models, except CNRM-CM6−1 are used to estimate the impacts of internal climate variability on the decadal change of WVB in the TP. Both model outputs and observations are re-gridded to a common resolution of $1.5° \times 1.5°$ using the bilinear interpolation method. In this study, we first calculate each model average with three ensemble members, which is then used to generate the multi-model mean.

## Calculation of WVB

In this study, we calculate the vertically integrated WVB at monthly time-scale based on Trenberth and Guillemot[49]:

$$P - E = -\frac{\partial \langle q \rangle}{\partial t} - \nabla \cdot \langle q\boldsymbol{V} \rangle + R \tag{1}$$

Where $P$, $E$, $q$, $\boldsymbol{V}$ and $R$ represent precipitation, evaporation, specific humidity, horizontal wind, and residual term, respectively. $\langle \ \rangle$ represents the vertical integration from the surface to top of atmosphere. The first term of $-\frac{\partial \langle q \rangle}{\partial t}$ represents the time change of atmospheric moisture storage, which can be negligible at monthly time-scale. The second term of $-\nabla \cdot \langle q\boldsymbol{V} \rangle$ represents the vertically integrated moisture flux divergence, which can be calculated through water vapor fluxes at four boundaries of the TP based on the 2D divergence theorem (Supplementary Fig. 9):

$$\iint -\nabla \cdot \langle q\boldsymbol{V} \rangle dA = \int -\langle q\boldsymbol{V} \rangle dl \tag{2}$$

Where $A$ represents the area of the TP; $l$ represents the length of boundaries. Here, we further calculate the net WVB across the boundaries of TP as follows:

$$\int -\langle q\boldsymbol{V} \rangle dl = B_W + B_S - B_N - B_E \tag{3}$$

Where $B_E$, $B_W$, $B_S$ and $B_N$ (kg s$^{-1}$) present the water vapor flux in eastern, western, southern, and northern boundaries, respectively, which can be calculated as follows:

$$\begin{cases} B_E = \int Q_u \, dl_E \\ B_W = \int Q_u \, dl_W \\ B_S = \int Q_v \, dl_S \\ B_N = \int Q_v \, dl_N \end{cases} \tag{4}$$

Where $Q_u$ (kg m$^{-1}$ s$^{-1}$) and $Q_v$ (kg m$^{-1}$ s$^{-1}$) represent the vertically integrated zonal and meridional water vapor flux, respectively, which can be calculated as follows:

$$\begin{cases} Q_u = -\frac{1}{g} \int_{p_s}^{p_t} qu \, dp \\ Q_v = -\frac{1}{g} \int_{p_s}^{p_t} qv \, dp \end{cases} \tag{5}$$

Where $g$ represents gravitational acceleration; $q$ represents specific humidity; $u$ represents zonal wind vector; $v$ represents meridional wind vector; $p$ represents atmospheric pressure; $p_s$ represents surface pressure; $p_t$ represents pressure at upper bound (setting to 50 hPa).

## Data availability

The monthly meteorological variables from the ERA5 reanalysis are open access at https://www.ecmwf.int/en/forecasts/datasets/reanalysis-datasets/era5. The multi-model outputs from historical, hist-aer, hist-GHG, hist-nat, and piControl experiments can be obtained at https://esgf-node.llnl.gov/search/cmip6/.

## Code availability

Data analysis in this study was conducted using open-source programming languages and software, including Climate Data Operators (CDO; https://code.mpimet.mpg.de/projects/cdo) and NCAR Command Language (NCL; http://www.ncl.ucar.edu/). All figures were produced using NCL.

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

## Acknowledgements

We thank the World Climate Research Program, which coordinated and facilitated CMIP6 through its Coupled Modelling Working Group, and thank Dr. Xiaochao Yu to help with plotting the schematic diagram. This work was supported by the National Natural Science Foundation of China (42341202, 41825011, 42275042, and 42205118) and the Science and Technology Development Fund of CAMS (2022KJ004).

## Author contributions

Z.W. and H.C. conceived the study. Z.W. and Y.L. contributed data analysis, figure generation, and writing. H.C., B.W. and X.Z. provided valuable comments and revised the study.

## Competing interests

The authors declare no competing interests.
