## [Peer Review File · Nature Communications]

Aerosol forcing regulating recent decadal change of summer water vapor budget over the Tibetan PlateauREVIEWER COMMENTS

Reviewer #1 (Remarks to the Author):

In this study, the authors address two important topics: climate change over the Tibetan Plateau (TP) and aerosol forcing. ERA5 reanalysis data and several CMIP6 models were used to identify the reason for the increased wetting of the TP in recent decades. It was found that a significant decrease in the flux of moisture from the eastern boundary of the TP is the main contributor to the recent increase in total column water vapor over TP. This decrease, in turn, is related to a circulation anomaly induced by asymmetric aerosol forcings over Europe and East Asia. The conclusions are interesting. But I hope they would have been more convincing. My foremost reason for worry is the methodology. The study uses reanalysis and model-derived data sets at pressure level over a complex topography where the data itself could be erroneous due to inter/extrapolation, and the effect of vertical flux cannot be ignored. Secondly, it is not shown if the moisture budget is closed. Without a closed budget, it is difficult to conclude if a particular component is responsible for an observed change. My detailed comments are below.

Major comments

1. The moisture budget at the time scale of a season or longer usually has very small components of the storage term (TPW). This insignificant term is at the center of this study. An increase in TPW with year due to flux, that is a couple of orders of magnitude higher than the $d(\text{TPW})/dt$ must be justified through a budget that is closed.
2. The study region is covered by complex topography. Over such regions. Moisture flux through mountain slopes is important. Can pressure level data capture such fluxes? Can it be shown that those fluxes are not important?
3. As a continuation of the previous comment, the role of evaporation is not described. Due to global warming, the retreat of glaciers and the increase of water bodies at that altitude could increase evaporation. What effect the surface evaporation have on the budget of moisture and its trend?
4. It is not obvious if the large-scale circulation anomaly (Fig 1a, 3) is due to east-west asymmetric aerosol forcing or due to a change in rainfall over the TP itself. A change in rainfall and related diabatic heating could change the circulation pattern and reduce the outgoing flux of moisture through the eastern boundary on account of increased convergence.

Minor comments:

1. It is not clear what the analysis is done for which season. Is it the annual average of the boreal summer average? It's not mentioned anywhere in the text (am I missing something?).

2. It is also not clear why the study is restricted from 1979 to 2014. The ERA5 data is available from 1940 up to recent time and a longer data set should provide more robust results, along with long CMIP6 runs.

3. In Fig 4 (as well as Supp Fig S6), it is mentioned that the data is averaged between 90-120E. That means this dimension is reduced. As yet, the x-axis ticks are longitude.

Reviewer #2 (Remarks to the Author):

The authors present an investigation into the driving causes behind decadal scale changes in the summertime water vapor budget over the Tibetan plateau. They use the standard driver attribution method of analysing the changes in relevant physical proxies in single forcer climate model simulations, and conclude that there is an outsized influence from the changes in anthropogenic aerosol emissions since 1979.

Tibetan Plateau moisture budgets is clearly an important topic, and understanding the underlying drivers of recent changes is crucial. Overall, the authors seem to have performed a thorough analysis, and reached a number of insightful conclusions. I do have a few methodological concerns, however. Also, importantly, the presentation of the results makes it difficult to fully follow the logic of the paper. I therefore recommend major revisions before this paper can be considered for publication.

Also, I note that a very similar paper appeared in Nature very recently. While the authors were likely unaware of this work, it would be important to include, discuss and compare to the results of Jiang et al. in the revised manuscript:

<https://www.nature.com/articles/s41586-023-06619-y>

Methodological points:

- L125: I am very surprised that the authors chose to exclude the two CMIP6 models that show a differing trend to observations. Since the objective of this study is to disentangle the effects of various drivers, these models would be key for understanding whether one particular driver has a strong influence. As an example, the high climate sensitivity of CanESM5 could be driving a strong moisture influence due to CO2 changes. Excluding models "to improve model confidence level" (L128) is not defensible in this case.

- The authors use three ensemble members from each model in the single forcer cases. I know this is at times all that is available, but for weak changes like aerosol influences on a regional scale it is clear that even three ensemble members may be too little. Is there one or more models where there are more members available, where the authors could test whether three members can be expected to give a reasonable representation of the forced signal?

- Would it be relevant to include a process schematic that illustrates what the different drivers do to moisture transport into/out of the TP, and why? This would, I think, make the dynamical explanations provided much easier to follow.

Presentation points:

- My main concern here is that so much of the key material resides in supplementary figures, that following the logic of the paper is virtually impossible without them. Also, the figures are not really discussed in order, but are brought into the discussion at various times and without clear explanation. I strongly encourage the authors to expand the number of figures in the main paper, and to slightly restructure the presentation so that it links closer to the graphics. Examples are Figures S4 and S5, which are key to understanding the forcing that would be driving the changes described.
- The current main figures also need units on the axes, and improved clarity. Notably figure 3 is more or less impossible to read, as the green arrows are hard to separate.

Minor points:

- L94: "A mount" -> "An amount"?
- L96: "Observations indicate" Which observations? (It's stated in the Methods, but please repeat it here for the benefit of the reader.)
- L114: This comment on another paper should perhaps be in the discussions?
- L143: "0.1 vs 0.13" These numbers seem identical, unless they have high precision. What is the uncertainty?
- L246: "see-saw like changes": By only using the one period, 1979-2014, the authors do not really investigate any see-saw like behaviour, I think? It underlies the results, but the trends shown would wash out any influence of a European emissions increase followed by a decrease?
- L261: "the problematic emissions inventory" Please explain what is meant here, rather than just referring to other studies.
- L262: "designing specific model experiments" These now exist:
<https://gmd.copernicus.org/articles/16/4451/2023/gmd-16-4451-2023.html>

We are grateful to the reviewers for your precious time in providing helpful comments and guidance that have improved the manuscript. In this document, we describe how we have addressed the reviewer's comments. Referee comments are shown in black and author responses are shown in blue.

Reviewer #1 (Remarks to the Author):

In this study, the authors address two important topics: climate change over the Tibetan Plateau (TP) and aerosol forcing. ERA5 reanalysis data and several CMIP6 models were used to identify the reason for the increased wetting of the TP in recent decades. It was found that a significant decrease in the flux of moisture from the eastern boundary of the TP is the main contributor to the recent increase in total column water vapor over TP. This decrease, in turn, is related to a circulation anomaly induced by asymmetric aerosol forcings over Europe and East Asia. The conclusions are interesting. But I hope they would have been more convincing. My foremost reason for worry is the methodology. The study uses reanalysis and model-derived data sets at pressure level over a complex topography where the data itself could be erroneous due to inter/extrapolation, and the effect of vertical flux cannot be ignored. Secondly, it is not shown if the moisture budget is closed. Without a closed budget, it is difficult to conclude if a particular component is responsible for an observed change. My detailed comments are below.

- Thank you very much for the positive and constructive comments and suggestions. All the questions and concerns have been carefully answered and the paper has been revised accordingly. For your concerns about complex topography impacts and closed moisture budget, please refer to our detailed responses shown below:

Major comments

1. The moisture budget at the time scale of a season or longer usually has very small components of the storage term (TPW). This insignificant term is at the center of this study. An increase in TPW with year due to flux, that is a couple of orders of magnitude higher than the $d(\text{TPW})/dt$ must be justified through a budget that is closed.

Response: Thanks for your comments! We are sorry that our Figure 1a (vertically integrated precipitable water vapor) may mislead you. We would like to clarify that we focused on moisture flux divergence term ($-\langle \nabla(q\mathbf{V}) \rangle$) not storage term ($-\langle \frac{\partial q}{\partial t} \rangle$) in this study. In the revised paper, we replaced Figure 1a with the difference ($P - E$) between precipitation and evaporation representing the net water entering a region and improved the descriptions of the methodology section as follows:

“Calculation of WVB

In this study, we calculate the vertically integrated WVB at monthly time-scale based on Trenberth and Guillemot ⁴⁹:

$$P - E = -\left\langle \frac{\partial q}{\partial t} \right\rangle - \langle \nabla(q\mathbf{V}) \rangle + R \quad (1)$$

Where P , E , q , \mathbf{V} and R represent precipitation, evaporation, specific humidity, horizontal wind and residual term, respectively. $\langle \rangle$ represents the vertical integration from the surface to top of atmosphere. The first term of $-\langle \frac{\partial q}{\partial t} \rangle$ represents the time change of atmospheric moisture storage, which can be negligible at monthly time-scale. The second term of $-\langle \nabla(q\mathbf{V}) \rangle$ represents the vertically integrated moisture flux divergence, which can be calculated through water vapor fluxes at four boundaries of the TP based on the 2D divergence theorem (Figure S10):

$$\iint -\langle \nabla(q\mathbf{V}) \rangle dA = \int -\langle q\mathbf{V} \rangle dl \quad (2)$$

Where A represents the area of the TP; l represents the length of boundaries. Here, we further calculate the net WVB across the boundaries of TP as follows:

$$\int -\langle q\mathbf{V} \rangle dl = B_W + B_S - B_N - B_E \quad (3)$$

Where B_E , B_W , B_S and B_N (kg s^{-1}) present the water vapor flux in eastern, western, southern and northern boundaries, respectively, which can be calculated as follows:

$$\begin{cases} B_E = \int Q_u dl_E \\ B_W = \int Q_u dl_W \\ B_S = \int Q_v dl_S \\ B_N = \int Q_v dl_N \end{cases} \quad (4)$$

Where Q_u ($\text{kg m}^{-1} \text{s}^{-1}$) and Q_v ($\text{kg m}^{-1} \text{s}^{-1}$) represent the vertically integrated zonal and meridional water vapor flux, respectively, which can be calculated as follows:

$$\begin{cases} Q_u = -\frac{1}{g} \int_{p_s}^{p_t} q u dp \\ Q_v = -\frac{1}{g} \int_{p_s}^{p_t} q v dp \end{cases} \quad (5)$$

Where g represents gravitational acceleration; q represents specific humidity; u represents zonal wind vector; v represents meridional wind vector; p represents atmospheric pressure; p_s represents surface pressure; p_t represents pressure at upper bound (setting to 50 hPa).” Lines 328-351

Figure 1. Changes of summer water vapor budget. (a) Spatial pattern of linear trends in the difference ($P - E$) between precipitation (P) and evaporation (E) (shade, unit: $\text{mm month}^{-1} \text{decade}^{-1}$) and water vapor flux (vector, unit: $\text{kg m}^{-1} \text{s}^{-1} \text{decade}^{-1}$) over the Tibetan Plateau during 1979-2014. The black points and green vectors indicate that the trends of precipitable water vapor and water vapor flux pass the test at 0.1 confidence level. **(b)** Time series (point) and linear fitting (line) of water vapor flux (unit: 10^6 kg s^{-1}) in four boundaries of the Tibetan Plateau during 1979-2014. The asterisks represent statistical significance at 90% confidence level. The numbers represent linear trends (unit: $10^6 \text{ kg s}^{-1} \text{decade}^{-1}$) of water vapor flux in each boundary of the Tibetan Plateau.

2. The study region is covered by complex topography. Over such regions. Moisture flux through mountain slopes is important. Can pressure level data capture such fluxes? Can it be shown that those fluxes are not important?

Response: Thanks for your comments! We agree that moisture flux through mountain slopes is important in complex topography. However, we would like to clarify that we diagnosed vertically integrated moisture flux divergence over the TP through

computing fluxes at the boundaries based on the 2D divergence theorem instead of centered finite differences for the rectilinear latitude-longitude grids (Please refer to the above reply for details). In four boundaries of the TP, the impacts of complex topography are limited. In the revised paper, we also clarified as follows:

“It is noted that previous studies revealed that complex topography can influence the diagnosis of moisture transport over the TP⁵⁰. However, there are limited impacts of complex topography on WVB in this study, because we select a line integral method based on the 2D divergence theorem to diagnose the net WVB over the TP.” Lines 352-355

3. As a continuation of the previous comment, the role of evaporation is not described. Due to global warming, the retreat of glaciers and the increase of water bodies at that altitude could increase evaporation. What effect the surface evaporation have on the budget of moisture and its trend?

Response: As responded to the first comment, our study mainly focused on the difference ($P - E$) between precipitation and evaporation. According to the suggestion, we have analyzed the trend of evaporation and added the descriptions as follows:

“The increase of WVB has greatly contributed to the wetting trend of the TP in recent decades^{22, 30, 31} (Figure S2c), although there is a slight increase in evaporation due to global warming³² (Figure S2b).” Lines 100-102

Figure S2 Observed linear trends of summer mean precipitation (P , **a**), evaporation (E , **b**) and its difference ($P - E$, **c**) during 1979-2014. The black points represent statistical significance at 90% confidence level.

4. It is not obvious if the large-scale circulation anomaly (Fig 1a, 3) is due to east-west asymmetric aerosol forcing or due to a change in rainfall over the TP itself. A change in rainfall and related diabatic heating could change the circulation pattern and reduce the outgoing flux of moisture through the eastern boundary on account of increased convergence.

Response: Thanks a lot for this constructive comment. As you noted, the diabatic heating associated with the variations in precipitation can modulate large-scale circulation anomalies. However, it is worth noting that in this study the circulation anomalies responsible for the precipitation anomalies belongs to a large-scale anomalous anticyclone centered to the northeast of the TP with distance more than 3000 km. It is difficult to attribute the formation of the anticyclone of this spatial scale to

heating anomalies associated with TP precipitation anomalies. Furthermore, both the anticyclone and TP precipitation anomalies are atmospheric anomalies, which cannot maintain longer than a month without external forcing factors. However, we also acknowledge that a change in rainfall over the TP caused by anomalous circulation and related diabatic heating release may in turn intensify the circulation anomaly. So, we added the descriptions in the revised paper as follows:

“Therefore, the decrease in water vapor export from the eastern boundary driven by atmospheric circulation anomaly dominates the increase in summer precipitation over the TP. The more latent heat release from increasing precipitation may strengthen convection development, which may in turn intensify anomalous anticyclone in the upper troposphere of the northern TP³⁹, thus forming a local positive feedback effect.”

Lines 192-197

Minor comments:

1. It is not clear what the analysis is done for which season. Is it the annual average of the boreal summer average? It's not mentioned anywhere in the text (am I missing something?).

Response: Our study focused on summer mean water vapor budget over the TP. It's mentioned in title, abstract and text in lines 82, 97, 105, 128, 135, 156, 167..... Additionally, we have added the “summer mean” in appropriate places of the text through careful checking.

2. It is also not clear why the study is restricted from 1979 to 2014. The ERA5 data is available from 1940 up to recent time and a longer data set should provide more robust results, along with long CMIP6 runs.

Response: Thanks for your comments! This study focused on the period of 1979-2014 due to the following reason:

The accuracy of reanalysis before 1979 remains to be debated, especially in areas with rare observations such as the Tibet Plateau due to the lack of satellite data assimilation. That's also why most reanalysis datasets (e.g., MERRA-2, NCEP-2 and JRA-55) started

in 1979.

3. In Fig 4 (as well as Supp Fig S6), it is mentioned that the data is averaged between 90-120E. That means this dimension is reduced. As yet, the x-axis ticks are longitude.

Response: Sorry for our mistakes in Fig.4 and Fig.S6. The x-axis ticks should be latitude and we have corrected it in the revised paper (new Figures 6 and S9).

Reviewer #2 (Remarks to the Author):

The authors present an investigation into the driving causes behind decadal scale changes in the summertime water vapor budget over the Tibetan plateau. They use the standard driver attribution method of analyzing the changes in relevant physical proxies in single forcer climate model simulations, and conclude that there is an outsized influence from the changes in anthropogenic aerosol emissions since 1979. Tibetan Plateau moisture budgets is clearly an important topic, and understanding the underlying drivers of recent changes is crucial. Overall, the authors seem to have performed a thorough analysis, and reached a number of insightful conclusions. I do have a few methodological concerns, however. Also, importantly, the presentation of the results makes it difficult to fully follow the logic of the paper. I therefore recommend major revisions before this paper can be considered for publication.

- Thank you very much for the positive and helpful evaluations. All the questions and concerns have been carefully answered and the paper has been revised accordingly.

Also, I note that a very similar paper appeared in Nature very recently. While the authors were likely unaware of this work, it would be important to include, discuss and compare to the results of Jiang et al. in the revised manuscript:

<https://www.nature.com/articles/s41586-023-06619-y>

Response: Accepted. We have compared and discussed the results of Jiang et al. (2023) as follows in the revised paper:

“Meanwhile, compared with a recent report by Jiang et al.⁴¹, which showed that the decrease in meridional troposphere temperature gradient over Eurasia caused by the uneven anthropogenic aerosol emissions led to the weakening of the Asian subtropical westerly jet, which ultimately dominated the increase in summer precipitation in the northern TP since the 1950s, we suggests a new mechanism, i.e., a teleconnection pathway across Eurasia induced by European aerosol forcing. We further reveal that the above mechanism and anomalous temperature gradient in eastern China due to the uneven aerosol forcing in Eurasia collectively play a dominant role in the weakening

of the westerly jet over the eastern TP. The weakened westerly winds decrease the water vapor output from the eastern boundary of the TP, rather than other boundaries, which may be the primary cause of increase in precipitation in the northern TP.” Lines 239-250

Methodological points:

- L125: I am very surprised that the authors chose to exclude the two CMIP6 models that show a differing trend to observations. Since the objective of this study is to disentangle the effects of various drivers, these models would be key for understanding whether one particular driver has a strong influence. As an example, the high climate sensitivity of CanESM5 could be driving a strong moisture influence due to CO₂ changes. Excluding models "to improve model confidence level" (L128) is not defensible in this case.

Response: Thanks for your comments!

(i) We agree that different models have different climate sensitivities to a specific forcing, e.g., greenhouse gases and aerosol. However, we first need to ensure consistency between historical simulations and observation for a specific study, e.g., the increasing water vapor flux in the eastern boundary of the TP in the past four decades. We usually consider the mechanism analysis to be reliable, only when the historical simulation for a specific model is consistent with observation. In this study, two climate models showed an opposite trend of water vapor flux compared to observation, which may be attributed to a lower climate sensitivity of aerosol and a higher climate sensitivity of greenhouse gases. Such climate model selection based on observation and historical simulations were also common in many previous studies.

(ii) We also analyzed the results of above two models and discussed in the revised paper as follows:

“We select nine climate models to further analyze the possible mechanisms behind the recent decadal change of summer WVB because CanESM5 and NorESM2-LM simulate a trend of water vapor flux in the eastern boundary of the TP contrary to the observation. However, it is noted that both CanESM5 and NorESM2-LM still show the

largest positive trend of summer WVB in the eastern boundary of the TP in the response to aerosol forcing (Figure S7).” Lines 257-262

Figure S7 Attribution of changes in summer water vapor budget in the eastern boundary of the Tibetan Plateau from NorESM2-LM and CanESM5. The gray, red, blue and green points represent simulated linear trends (unit: $10^6 \text{ kg s}^{-1} \text{ decade}^{-1}$) of water vapor budget in eastern boundary of the Tibetan Plateau during 1979-2014 from all (ALL), aerosol-only (AER), greenhouse gas-only (GHG) and natural-only (NAT) forcings simulations, respectively.

- The authors use three ensemble members from each model in the single forcer cases. I know this is at times all that is available, but for weak changes like aerosol influences on a regional scale it is clear that even three ensemble members may be too little. Is there one or more models where there are more members available, where the authors could test whether three members can be expected to give a reasonable representation of the forced signal?

Response: As suggested, we added the new Figure S6 to compare the results of ten ensemble members from IPSL-CM6A-LR and MIROC6. The conclusion that aerosol forcing regulate the recent decadal change of summer water vapor budget over the Tibetan Plateau remains the same in the results from ten ensemble members from two models. Also, we added these descriptions in the revised paper:

“Although the only first three ensemble members in each model are used in this study, the main conclusion that aerosol forcing regulate the recent decadal change of summer

water vapor budget over the Tibetan Plateau remains the same in ten ensemble members from IPSL-CM6A-LR and MIROC6 (Figure S6).” Lines 254-257

Figure S6 Attribution of changes in summer water vapor budget in the eastern boundary of the Tibetan Plateau from ten ensemble members from IPSL-CM6A-LR and MIROC6. The gray, red, blue and green points represent simulated linear trends (unit: $10^6 \text{ kg s}^{-1} \text{ decade}^{-1}$) of water vapor budget in eastern boundary of the Tibetan Plateau during 1979-2014 from all (ALL), aerosol-only (AER), greenhouse gas-only (GHG) and natural-only (NAT) forcings simulations, respectively. The error bars represent one standard deviation of ten ensemble members for each model.

- Would it be relevant to include a process schematic that illustrates what the different drivers do to moisture transport into/out of the TP, and why? This would, I think, make the dynamical explanations provided much easier to follow.

Response: As suggested, we added the schematic diagram Figure 7 to illustrating the paths of uneven aerosol forcing in Eurasia regulating the recent decadal change of summer water vapor budget over the TP in the revised paper.

Figure 7. Schematic diagram illustrating the paths of uneven aerosol forcing in Eurasia regulating the recent decadal change of summer water vapor budget over the TP.

Presentation points:

- My main concern here is that so much of the key material resides in supplementary figures, that following the logic of the paper is virtually impossible without them. Also, the figures are not really discussed in order, but are brought into the discussion at various times and without clear explanation. I strongly encourage the authors to expand the number of figures in the main paper, and to slightly restructure the presentation so that it links closer to the graphics. Examples are Figures S4 and S5, which are key to understanding the forcing that would be driving the changes described.

Response: As suggested, we removed some key figures from Supplementary to main text, including Figure S2, Figure S4 and Figure S5. Additionally, we adjusted the order of some tables and figures to read logically.

- The current main figures also need units on the axes, and improved clarity. Notably figure 3 is more or less impossible to read, as the green arrows are hard to separate.

Response: As suggested, we added the units on the axes for all figures in main text and Supplementary. Additionally, we revised the figure 3 (new figure 4) to show arrows clearly.

Minor points:

- L94: "A mount" -> "An amount"?

Response: Corrected.

- L96: "Observations indicate" Which observations? (It's stated in the Methods, but please repeat it here for the benefit of the reader.)

Response: As suggested, we revised "observations" as "The ERA5 reanalysis" Line 96

- L114: TH is comment on another paper should perhaps be in the discussions?

Response: As suggested, we removed this comment to discussion section. Lines 235-239

- L143: "0.1 vs 0.13" These numbers seem identical, unless they have high precision. What is the uncertainty?

Response: As suggested, we revised the units from $10^6 \text{ kg s}^{-1} \text{ year}^{-1}$ to $10^6 \text{ kg s}^{-1} \text{ decade}^{-1}$. This comparison has changed to "1.0 vs 1.3" in the revised paper. Line 140

- L246: "see-saw like changes": By only using the one period, 1979-2014, the authors do not really investigate any see-saw like behaviour, I think? It underlies the results, but the trends shown would wash out any influence of a european emissions increase followed by a decrease?

Response: In this study, "see-saw like" represents spatial pattern (decreasing anthropogenic emission in Europe but increasing anthropogenic emission in China) not time variation. To avoid misleading readers, we have revised "see-saw like aerosol forcing" to "uneven or inhomogeneous aerosol forcing".

- L261: "the problematic emissions inventory" Please explain what is meant here, rather than just referring to other studies.

Response: As suggested, we clarified as follows in the revised paper:

"However, the CMIP6 models fail to capture the trends in AOD and SSR in China during this time due to an opposite trend of anthropogenic aerosol emissions in the Community Emissions Data System (CEDS) inventory^{42, 43}." Lines 288-291

- L262: "designing specific model experiments" These now exist: <https://gmd.copernicus.org/articles/16/4451/2023/gmd-16-4451-2023.html>

Response: Thanks very much for this useful information. We have added the descriptions as follows in the revised paper:

"It will be worthwhile to quantify the impact of reductions in aerosol emissions over China in the future by using specific model experiments such as the recent Regional

Aerosol Model Intercomparison Project (RAMIP) ⁴⁴.” Lines 291-294

REVIEWER COMMENTS

Reviewer #1 (Remarks to the Author):

The authors have made several changes to the manuscript in response to both reviewers. However, I still feel that some of the major concerns raised by me are still unresolved.

1. My first concern (Comment 1) was on the relative magnitude (contribution) of the tendency term of the moisture budget equation. In response, the authors have modified the analysis to include the P-E analysis instead of the advection of moisture. However, this does not answer my query as (P-E) includes $\langle dq/dt \rangle$ in addition to the advection term. If this tendency term is large, how can it be concluded that advection is responsible for changes in P-E? In summary, without closing the moisture budget, it is a bit difficult to make a profound statement on the causality of changes in P-E.

2. In addition, the equation of moisture budget needs to be corrected (in both the main and reply to the reviewers). The term $\langle \nabla(qV) \rangle$ is missing the dot product. Please correct.

3. I'm not sure how one can confirm that the effect of boundary slopes is limited in these calculations of gradient (in response to my Comment 2). Can it be shown, at least quoting some previous study?

Reviewer #2 (Remarks to the Author):

I thank the authors for their thorough revision. The updated manuscript is now much clearer, and I appreciate the addition of the schematic.

While I am broadly convinced by the arguments presented, I still have some concerns regarding the choice of models and ensemble members. The new figures S6 and S7 help, but still do not provide sufficient information to assure that the attribution presented in Figure 2 - which is key for the conclusions of the paper - is fully supported.

An example of the issue is the CanESM5 result in Figure S7. Its net flux is very close to zero, however the authors have chosen to exclude the model since its overall flux is negative. The results for the individual ensemble members are not shown, but it is very likely that at least one of them is net positive - and should therefore, by the authors methodology, have been included if the analysis was one on a per-member basis.

Excluding models based on the ensemble mean could still be defensible, however Figure S6, which shows 10-member results for two models, highlights the issue. Here, both models have a positive net, but with a 1 std.dev. spread that touches zero. Hence, there will be members in each sample that are negative. Consequently, it is fully possible to select a 3-member set that gives a negative mean flux, even

for a model where the 10-member mean is positive.

So, how can the authors be certain that this is not the case for the two excluded models? The answer is that they can't, at least based on a 3-member mean.

The easiest solution here is to include all models in the main result. Based on figure S6, this is unlikely to affect the overall conclusions, but it will give a less biased representation of errors. At the same time, I urge the authors to include the (3-member mean) results for all individual models in Figure 2 (or in a corresponding supplement). This gives a much clearer indication of how robust the modelling/attribution part of the study is.

If the authors still wish to exclude two models, I believe that a more thorough analysis is needed to show that these two really have robustly different behavior to the others.

Another suggestion that would help make the argument clearer is to also show the results for all members in figure S7. Standard deviations do not fully show the spread of the members, or how robust the mean behavior is.

I thank the authors again for their continued efforts, and hope that these final comments are taken as constructive.

We are grateful to the reviewers for your precious time in providing further helpful comments and guidance that have improved the manuscript. In this document, we describe how we have addressed the reviewer's comments. Referee comments are shown in black and author responses are shown in blue.

Reviewer #1 (Remarks to the Author):

The authors have made several changes to the manuscript in response to both reviewers. However, I still feel that some of the major concerns raised by me are still unresolved.

- Thanks for your positive evaluations for our revised version.

1. My first concern (Comment 1) was on the relative magnitude (contribution) of the tendency term of the moisture budget equation. In response, the authors have modified the analysis to include the P-E analysis instead of the advection of moisture. However, this does not answer my query as (P-E) includes $\langle dq/dt \rangle$ in addition to the advection term. If this tendency term is large, how can it be concluded that advection is responsible for changes in P-E? In summary, without closing the moisture budget, it is a bit difficult to make a profound statement on the causality of changes in P-E.

Response: Thanks for your further comments! As suggested, we compared relative magnitude of linear trends between atmospheric moisture storage and moisture flux divergence. The related contents are added in revised paper as follows:

“The increasing summer WVB over the TP can be attributed to the changes of atmospheric moisture storage and moisture flux divergence (Details in Equation 1 in Method section). During 1979-2014, the vertically integrated moisture flux divergence shows a significant increase ($p < 0.1$), with a seasonal average rate of $4.7 \text{ kg s}^{-1} \text{ decade}^{-1}$. In contrast, there are limited changes of atmospheric moisture storage ($-\frac{\partial \langle q \rangle}{\partial t}$), with a seasonal average rate of $-0.05 \text{ kg s}^{-1} \text{ decade}^{-1}$. This comparison demonstrates the dominated contribution of moisture flux divergence to increasing WVB over the TP.”

Lines 104-110

2. In addition, the equation of moisture budget needs to be corrected (in both the main and reply to the reviewers). The term $-\nabla \cdot \langle q\mathbf{V} \rangle$ is missing the dot product. Please correct.

Response: Thanks! The term $-\nabla \langle q\mathbf{V} \rangle$ has been corrected as $-\nabla \cdot \langle q\mathbf{V} \rangle$ in the revised paper.

3. I'm not sure how one can confirm that the effect of boundary slopes is limited in these calculations of gradient (in response to my Comment 2). Can it be shown, at least quoting some previous study?

Response: Thanks for your further comments! In previous versions, we used the simplified moisture flux divergence term $-\langle \nabla \cdot (q\mathbf{V}) \rangle$ instead of $-\nabla \cdot \langle q\mathbf{V} \rangle$. Based on the Equation 5b in Trenberth and Guillemot (1995), $-\nabla \cdot \langle q\mathbf{V} \rangle$ can be written as $-\nabla \cdot \langle q\mathbf{V} \rangle = -\langle \nabla \cdot (q\mathbf{V}) \rangle - q_s V_s \cdot \nabla p_s$. The second term $-q_s V_s \cdot \nabla p_s$ represents the effects of complex topography. As suggested, we calculated the effects of complex topography on the trend of moisture flux divergence (Figure R1). The results show that complex topography has a small effect on increasing moisture flux divergence over the TP during 1979-2014.

Furthermore, we calculated moisture flux divergence term through water vapor fluxes at four boundaries of the TP based on 2D divergence theorem $\iint \text{div } \mathbf{F} dA = \int \mathbf{F} dl$, that is $\iint -\nabla \cdot \langle q\mathbf{V} \rangle dA = \int -\langle q\mathbf{V} \rangle dl$ (Equation 2 in Main text). Indeed, the effects of complex topography are included in calculation of moisture flux divergence term through water vapor fluxes at four boundaries.

Reference: Trenberth, K. E. & Guillemot C. J. Evaluation of the global atmospheric moisture budget as seen from analyses. *Journal of Climate* 8, 2255-2272 (1995).

Figure R1 The panels (a), (b) and (c) represent linear trends of $-\nabla \cdot \langle qV \rangle$, $-\langle \nabla \cdot (qV) \rangle$ and $-q_s V_s \cdot \nabla p_s$ during 1979-2014, respectively.

Reviewer #2 (Remarks to the Author):

I thank the authors for their thorough revision. The updated manuscript is now much clearer, and I appreciate the addition of the schematic.

- Thanks for your positive evaluations for our revised version.

1. While I am broadly convinced by the arguments presented, I still have some concerns regarding the choice of models and ensemble members. The new figures S6 and S7 help, but still do not provide sufficient information to assure that the attribution presented in Figure 2 - which is key for the conclusions of the paper - is fully supported. An example of the issue is the CanESM5 result in Figure S7. Its net flux is very close to zero, however the authors have chosen to exclude the model since its overall flux is negative. The results for the individual ensemble members are not shown, but it is very likely that at least one of them is net positive - and should therefore, by the authors methodology, have been included if the analysis was one on a per-member basis.

Excluding models based on the ensemble mean could still be defensible, however Figure S6, which shows 10-member results for two models, highlights the issue. Here, both models have a positive net, but with a 1 std.dev. spread that touches zero. Hence, there will be members in each sample that are negative. Consequently, it is fully possible to select a 3-member set that gives a negative mean flux, even for a model where the 10-member mean is positive. So, how can the authors be certain that this is not the case for the two excluded models? The answer is that they can't, at least based on a 3-member mean. The easiest solution here is to include all models in the main result. Based on figure S6, this is unlikely to affect the overall conclusions, but it will give a less biased representation of errors. At the same time, I urge the authors to include the (3-member mean) results for all individual models in Figure 2 (or in a corresponding supplement). This gives a much clearer indication of how robust the modelling/attribution part of the study is. If the authors still wish to exclude two models, I believe that a more thorough analysis is needed to show that these two really have robustly different behavior to the others.

Response: We really agreed the further comments from Reviewer. As suggested, we used the simulations from all climate models to attribute and analyze the possible mechanisms behind increasing water vapor budget over the TP during 1979-2014. We have carefully updated the figures and related texts in the revised paper. Furthermore, following suggestions, we have revised the Figure 2 and included the results from all models in the revised paper as follows:

Figure 2. Attribution of changes in summer water vapor flux in the eastern boundary of the Tibetan Plateau. The gray, red, blue, green and orange markers represent simulated linear trends (unit: 10⁶ kg s⁻¹ decade⁻¹) of water vapor flux in eastern boundary of the Tibetan Plateau during 1979-2014 from all (ALL), aerosol-only (AER), greenhouse gas-only (GHG), natural-only (NAT) forcings and piControl (PI) simulations in eleven climate models, respectively. The points represent the ensemble means of eleven climate models. The error bars represent one standard deviation of eleven climate models.

2. Another suggestion that would help make the argument clearer is to also show the results for all members in figure S7. Standard deviations do not fully show the spread of the members, or how robust the mean behavior is.

Response: As suggested, we added the results for all members in Figure S7 (New Figure S6) in the revised paper as follows:

Figure S6 Attribution of changes in summer water vapor flux in the eastern boundary of the Tibetan Plateau from ten ensemble members from IPSL-CM6A-LR and MIROC6. The gray, red, blue and green markers represent simulated linear trends (unit: $10^6 \text{ kg s}^{-1} \text{ decade}^{-1}$) of water vapor flux in eastern boundary of the Tibetan Plateau during 1979-2014 from all (ALL), aerosol-only (AER), greenhouse gas-only (GHG) and natural-only (NAT) forcings simulations in ten ensemble members, respectively. The points and error bars represent ensemble means and one standard deviations of ten ensemble members for each model.

I thank the authors again for their continued efforts, and hope that these final comments are taken as constructive.

- Thanks for your constructive comments again.

REVIEWERS' COMMENTS

Reviewer #1 (Remarks to the Author):

The authors have provided reasonable responses to the second round of comments by both reviewers. At this point, I don't have any new queries and I congratulate the authors for a great job done.

Reviewer #2 (Remarks to the Author):

I thank the authors for this second revision. My concerns have been addressed in the revised figures, so I have no further comments.